# Replication Study: Melanoma genome sequencing reveals frequent *PREX2* mutations

**Stephen K Horrigan[1], Pascal Courville[2], Darryl Sampey[2], Faren Zhou[3], Steve Cai[3], Reproducibility Project: Cancer Biology***

[1]Noble Life Sciences, Sykesville, United States; [2]BioFactura, Rockville, United States; [3]TACGen, Richmond, United States

*For correspondence: tim@cos.io; nicole@scienceexchange.com

**Group author details:**
Reproducibility Project: Cancer Biology See page 13

**Abstract** In 2015, as part of the Reproducibility Project: Cancer Biology, we published a Registered Report (Chroscinski et al., 2014) that described how we intended to replicate selected experiments from the paper "Melanoma genome sequencing reveals frequent *PREX2* mutations" (Berger et al., 2012). Here we report the results of those experiments. We regenerated cells stably expressing ectopic wild-type and mutant phosphatidylinositol-3,4,5-trisphosphate-dependent Rac exchange factor 2 (PREX2) using the same immortalized human NRAS$^{G12D}$ melanocytes as the original study. Evaluation of PREX2 expression in these newly generated stable cells revealed varying levels of expression among the PREX2 isoforms, which was also observed in the stable cells made in the original study (Figure S6A; Berger et al., 2012). Additionally, ectopically expressed PREX2 was found to be at least 5 times above endogenous PREX2 expression. The monitoring of tumor formation of these stable cells *in vivo* resulted in no statistically significant difference in tumor-free survival driven by *PREX2* variants, whereas the original study reported that these *PREX2* mutations increased the rate of tumor incidence compared to controls (Figure 3B and S6B; Berger et al., 2012). Surprisingly, the median tumor-free survival was 1 week in this replication attempt, while 70% of the control mice were reported to be tumor-free after 9 weeks in the original study. The rapid tumor onset observed in this replication attempt, compared to the original study, makes the detection of accelerated tumor growth in *PREX2* expressing NRAS$^{G12D}$ melanocytes extremely difficult. Finally, we report meta-analyses for each result.

## Introduction

The Reproducibility Project: Cancer Biology (RP:CB) is a collaboration between the Center for Open Science and Science Exchange that seeks to address concerns about reproducibility in scientific research by conducting replications of selected experiments from a number of high-profile papers in the field of cancer biology (*Errington et al., 2014*). For each of these papers a Registered Report detailing the proposed experimental designs and protocols for the replications was peer reviewed and published prior to data collection. The present paper is a Replication Study that reports the results of the replication experiments detailed in the Registered Report (*Chroscinski et al., 2014*) for a paper by Berger et al., and uses a number of approaches to compare the outcomes of the original experiments and the replications.

In 2012, Berger et al. sequenced the whole genome of 25 metastatic melanomas and matched germline DNA. The authors identified the phosphatidylinositol-3,4,5-trisphosphate-dependent Rac exchange factor 2 gene (*PREX2*) as a significantly mutated gene (SMG) in this population and went on to confirm this finding in an independent cohort of 107 human melanoma samples. To investigate the functional relevance of *PREX2* mutations, six of the identified mutant PREX2 isoforms

were ectopically expressed in immortalized human melanocytes and tumor formation was monitored after injecting into immunodeficient mice. Four of the mutations, three truncating variants as well as a point mutant, resulted in a statistically significant decrease in tumor-free survival compared to control melanocytes, expressing wild-type PREX2 (PREX2$^{WT}$) or green fluorescent protein (GFP).

The Registered Report for the paper by Berger et al. described the experiments to be replicated (Figures 3B and S6), and summarized the current evidence for these findings (*Chroscinski et al., 2014*). While *Berger et al. (2012)* reported *PREX2* as an SMG in melanoma, other studies have failed to identify *PREX2* as an SMG in genome-wide screens of melanoma samples (*Cancer Genome Atlas Network, 2015*; *Hodis et al., 2012*; *Krauthammer et al., 2012*; *Marzese et al., 2014*; *Ni et al., 2013*), including a meta-analysis of over 200 samples (*Xia et al., 2014*). Recently, *PREX2* was identified as an SMG in pancreatic cancer samples using a whole-genome approach with a mutation rate of ~10% (*Waddell et al., 2015*), similar to the reported rate in *Berger et al. (2012)*. Further, one of the truncating mutations specific to melanocytes (PREX2$^{E824*}$) identified in *Berger et al. (2012)* was further explored to determine the *in vivo* implications of this mutation in the context of mutant NRAS. Although the PREX2$^{E824*}$ mutation was not included in this replication attempt, Lissanu Deribe and colleagues reported that a genetically engineered conditional knockout mouse harboring the *Prex2$^{E824*}$* mutation accelerated melanoma development compared to control mice (*Lissanu Deribe et al., 2016*).

The outcome measures reported in this Replication Study will be aggregated with those from the other Replication Studies to create a dataset that will be examined to provide evidence about reproducibility of cancer biology research, and to identify factors that influence reproducibility more generally.

## Results and discussion

### Sequencing of endogenous *PREX2* in NRAS$^{G12D}$ melanocytes

Using the same TERT-immortalized human melanocytes engineered to express NRAS$^{G12D}$ (NRAS$^{G12D}$ melanocytes) as the original study, we determined the genetic status of the endogenous *PREX2* gene. This was not included in the original study (*Berger et al., 2012*); however, was suggested during peer review of the Registered Report to understand if the genetic background of the cell line might influence the interpretation of study results. We generated PCR products which covered the coding region of *PREX2* and generated DNA sequence using the Sanger method (*Sanger et al., 1977*). Ultimately, we achieved an average of 4.5x coverage for bases contained within the coding region of the *PREX2* gene (RefSeq: NM_024870.3, GRCh38/hg38 Assembly), which gave sufficient confidence in the base called at each position (*Figure 1*). No severe coding or splice site mutations were detected; however, four coding single nucleotide polymorphisms (SNPs) and 1 5'UTR SNP were identified (*Figure 1—figure supplement 1*).

### Confirming ectopic expression of PREX2 mutant isoforms by Western blot

For this replication attempt we regenerated NRAS$^{G12D}$ melanocytes harboring stable integration of GFP, PREX2$^{WT}$, PREX2$^{G844D}$, or PREX2$^{Q1430*}$ isoforms. This experiment is similar to what was reported in Figure S6 of *Berger et al. (2012)*. The PREX2$^{G844D}$ isoform contains a missense variant in exon 22 of the *PREX2* gene (c.2531G>A) and the PREX2$^{Q1430*}$ isoform contains a nonsense truncating variant in exon 35 of the *PREX2* gene (c.4288C>T). Although Berger et al. tested six different isoforms in total, PREX2$^{G844D}$ and PREX2$^{Q1430*}$ isoforms were selected for evaluation because they each had the greatest effect on tumor free survival within their respective mutation classes (i.e., missense and nonsense).

Expression of ectopic PREX2 isoforms was evaluated by Western blot through detection of the V5 tag, similar to the original study. Relative expression levels (V5/Tubulin) for the PREX2$^{G844D}$ and PREX2$^{Q1430*}$ isoforms made during this replication attempt were 0.6 and 1.7 times, respectively, the amount of PREX2$^{WT}$ protein ectopically expressed. This compares to 0.6 and 0.7 times for the PREX2$^{G844D}$ and PREX2$^{Q1430*}$ isoforms reported in *Berger et al. (2012)*. We also re-analyzed the relative expression levels in the stable NRAS$^{G12D}$ melanocytes generated in the original study, which were shared by the original authors. Western blot analysis resulted in a relative expression of ~0.5

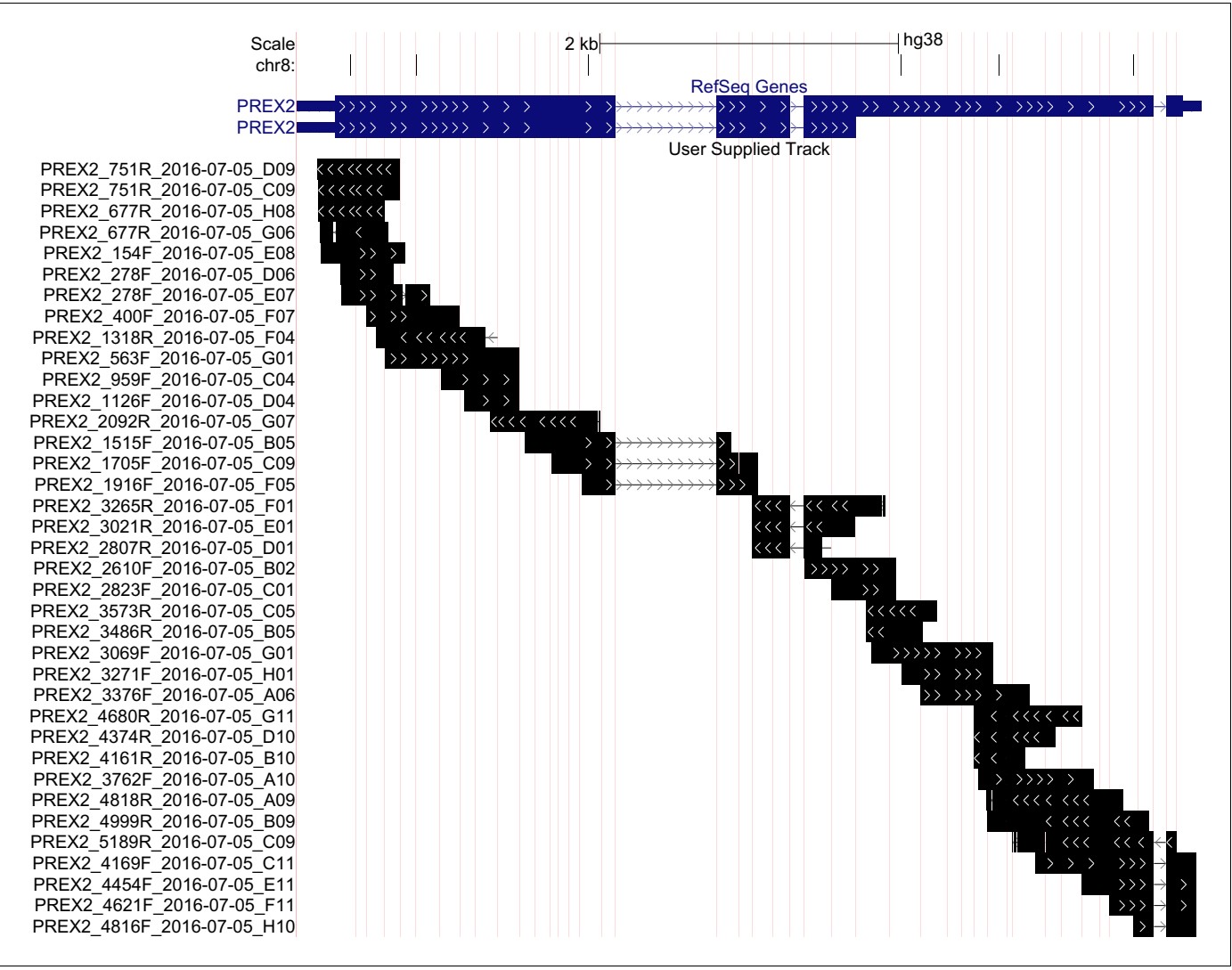

**Figure 1.** Sequencing of endogenous *PREX2* gene in NRAS[G12D] melanocytes. Representation of sequencing coverage of *PREX2* by Sanger sequencing. Reference gene isoforms shown in blue, sequence coverage seen in black, and sequencing run name shown on the y-axis. Vertical red lines represent exon/intron breaks. White arrows indicate strand of the sequence. Image created using UCSC Genome Browser's custom track feature in multi-region, exon only view, GRch38/hg38 (http://goo.gl/JLezy5). Additional information can be found at https://osf.io/r53z8/.

The following figure supplement is available for figure 1:

**Figure supplement 1.** Endogenous *PREX2* sequence evaluation.

times for both the PREX2[G844D] and PREX2[Q1430*] isoforms compared to PREX2[WT] protein. Not unexpectedly, PREX2[WT] protein levels also varied between the original and replication stable cells, with the stable cells generated in the original study expressing over two times as much protein compared to the PREX2[WT] stable cells generated in this replication attempt. However, as seen in *Figure 2B*, the mean expression of all PREX2 isoforms made during this replication attempt fall within the 95% CI of those generated during the original study and are similar to those reported in the original manuscript.

To test if ectopic PREX2 expression was different among the stable cells, an analysis of variance (ANOVA) was performed. The ANOVA had three levels of PREX2-V5 isoform expression in NRAS[G12D] melanocytes (PREX2[WT], PREX2[G844D], or PREX2[Q1430*]) and two levels of study (generated during this replication attempt or the original study). The ANOVA result for the main effect of

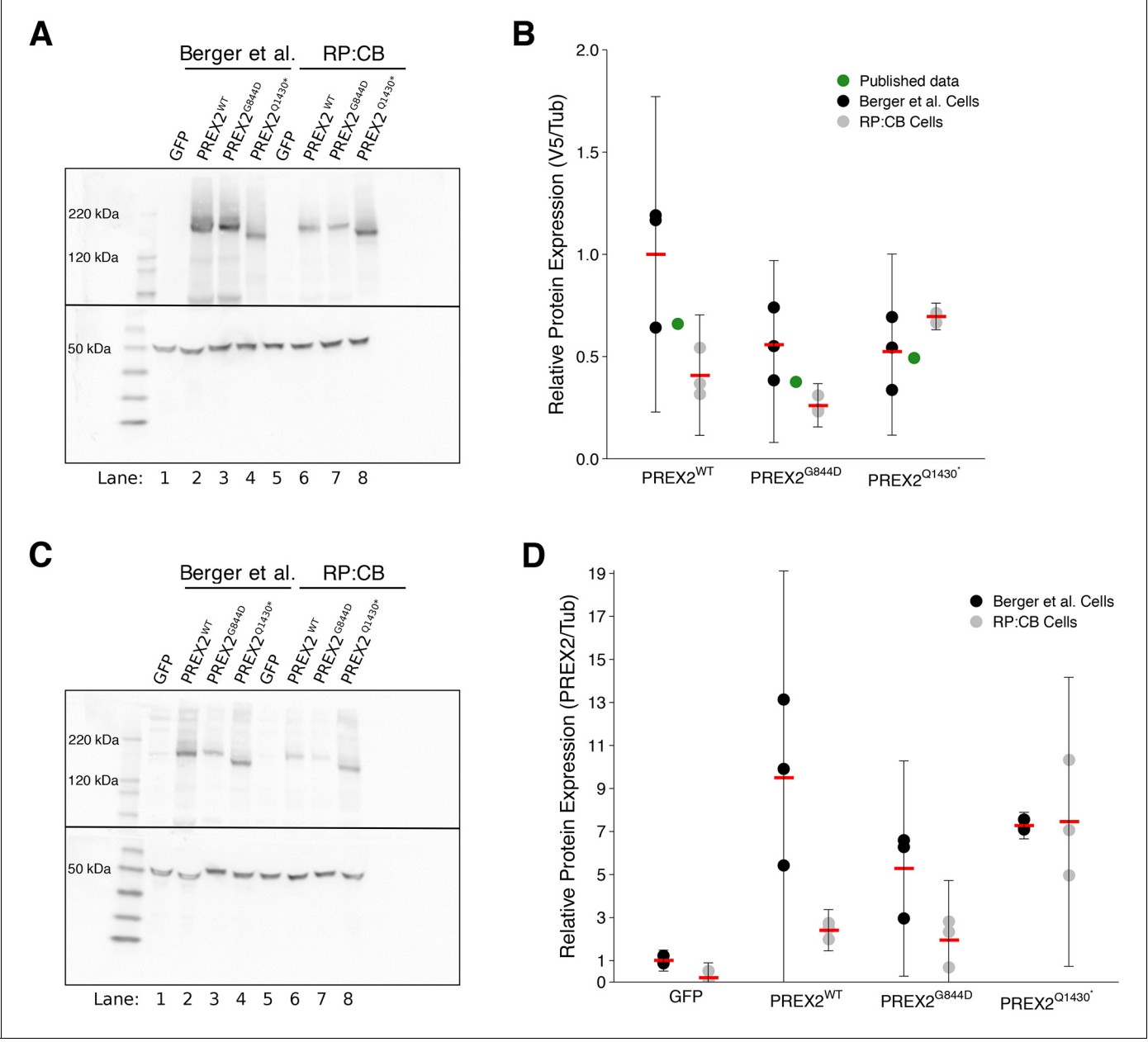

**Figure 2.** Expression of PREX2 isoforms. NRAS$^{G12D}$ melanocytes were transduced to express GFP or PREX2 isoforms (PREX2$^{WT}$, PREX2$^{G844D}$, PREX2$^{Q1430*}$). (**A**) Representative Western blot using an anti-V5 antibody (top panel) or an anti-$\alpha$-Tubulin antibody (bottom panel). Lanes 1–4 are from cells transduced with GFP, PREX2$^{WT}$, PREX2$^{G844D}$, or PREX2$^{Q1430*}$, respectively, generated during the original study (*Berger et al., 2012*). Lanes 5–8 are from cells transduced with GFP, PREX2$^{WT}$, PREX2$^{G844D}$, or PREX2$^{Q1430*}$, respectively, made during this replication attempt (RP:CB). Membranes were cut at ~70 kDa so that V5 and $\alpha$-Tubulin could be probed in parallel. (**B**) Relative protein expression (V5/Tubulin) of ectopically expressed PREX2 isoforms are presented for each stable cell line and normalized to PREX2$^{WT}$ cells made during the original study. Means represented by red bars, circles indicate individual biological replicates [n = 3], and error bars indicate 95% CI. Green circles represent protein expression reported in Figure S6A of the original study. Two-way ANOVA main effect of PREX2 isoform($F_{(2,12)}$ = 4.608, $p$=0.033), the main effect of study ($F_{(1,12)}$ = 8.701, $p$=0.012), and isoform:study interaction ($F_{(2,12)}$ = 7.52, $p$=0.008). (**C**) Representative Western blot using an anti-PREX2 antibody (top panel) or an anti-$\alpha$-Tubulin antibody (bottom panel). Lanes 1–4 are from cells generated during the original study transduced with GFP, PREX2$^{WT}$, PREX2$^{G844D}$, or PREX2$^{Q1430*}$, respectively. Lanes 5–8 are from cells made during this replication attempt transduced with GFP, PREX2$^{WT}$, PREX2$^{G844D}$, or PREX2$^{Q1430*}$, respectively. Membranes were cut at ~70 kDa so that PREX2 and $\alpha$-Tubulin could be probed in parallel. (**D**) Relative protein expression (PREX2/Tubulin) of ectopically expressed PREX2 isoforms are presented for each stable cell line and normalized to PREX2$^{WT}$ cells generated during the original study. Means represented by red bars, circles indicate individual biological replicates [n = 3], and error bars indicate 95% CI. Two-way ANOVA main effect of PREX2 expression ($F_{(3,16)}$ = 15.033, $p$=6.47$\times10^{-5}$), the main effect of study ($F_{(1,16)}$ = 13.08, $p$=0.002), expression:study interaction ($F_{(3,16)}$ = 4.50, $p$=0.018). Additional details for this experiment can be found at https://osf.io/4c8tu/.

PREX2 isoform ($F(2,12) = 4.608$, $p=0.033$), the main effect of study ($F(1,12) = 8.701$, $p=0.012$), as well as the PREX2 isoform:study interaction effect ($F(2,12) = 7.525$, $p=0.008$) were statistically significant. While this indicates the null hypothesis that any differences between isoform expression is the same for each study, and vice versa, can be rejected, it is difficult to interpret because the main effects and interaction effect are all statistically significant. If desired, multiple comparison post-hoc tests could be done to explore how expression differs, but this result does suggest that isoform expression is not equal, which is of particular interest for this study.

While the V5 tag allows for easy detection of ectopically expressed PREX2 isoforms, it does not provide a means to understand how the ectopic expression compares to endogenous PREX2. This aspect was not examined in the original study. The relative expression of endogenous and ectopic PREX2 was evaluated by Western blot using an anti-PREX2 antibody which recognizes amino acids 960–973 (*Hodakoski et al., 2014*) (*Figure 2C–D*). Relative expression (PREX2/Tubulin) for PREX2$^{WT}$ made during this replication attempt was 12 times the endogenous PREX2 level detectable in GFP stable cells, while the PREX2$^{G844D}$ and PREX2$^{Q1430*}$ isoforms were 10 and 38 times the endogenous levels, respectively. Comparatively, the stable PREX2$^{WT}$, PREX2$^{G844D}$, and PREX2$^{Q1430*}$ cells generated during the original study were 9, 5, and 7 times over detectable endogenous PREX2, respectively. However, similar to the V5 analysis, the mean expression of all PREX2 isoforms, aside from GFP, in cells made during this replication attempt fall within the 95% CI of those generated during the original study (*Figure 2D*).

An ANOVA was performed to test if PREX2 expression was similar in NRAS$^{G12D}$ melanocytes across the different isoforms as well as between the originally generated stable cells and the stable cells generated in this replication attempt. This ANOVA had four levels of PREX2 expression (GFP, PREX2$^{WT}$, PREX2$^{G844D}$, or PREX2$^{Q1430*}$) and two levels of study (generated during this replication attempt or the original study). The ANOVA result for the main effect of PREX2 expression ($F(3,16) = 15.033$, $p=6.47\times10^{-5}$), the main effect of study ($F(1,16) = 13.08$, $p=0.002$), as well as the interaction effect ($F(3,16) = 4.499$, $p=0.018$) were statistically significant. As with the ANOVA results on the V5 expression data, these results are difficult to interpret, but it does suggest that expression is not the same. Any dissimilarity in expression between the stable cells could be due to a number of factors, such as differences in integration site and copy number, differences in growth conditions between laboratories, and the inherent variability of the Western blot technique (*Koller and Wätzig, 2005*).

## Generation of tumor xenografts expressing PREX2 isoforms and evaluation of tumor-free survival and tumor growth

We sought to independently replicate if the expression of different isoforms of PREX2 impacted tumor-free survival (TFS) *in vivo*. Female athymic nude mice were subcutaneously injected with the human NRAS$^{G12D}$ melanocytes generated during this replication attempt, harboring *PREX2* variants or *GFP*, with the intention of monitoring TFS for 16 weeks. The number of mice and length of monitoring were determined *a priori* to have sufficient power to detect the originally reported effect sizes (*Chroscinski et al., 2014*). This experiment is similar to what was reported in Figure 3B and S6B of *Berger et al. (2012)*. Following injection of the different melanocytic lines, tumor incidence was detected in all mice within three weeks. Median TFS, defined as the time at which at least 50% of the mice had a palpable tumor, was one week for each of the four groups of mice (*Figure 3A*). This compares to the original study which reported the median TFS at five weeks for mice injected with PREX2$^{G844D}$ and PREX2$^{Q1430*}$ expressing melanocytes. This was significantly less than PREX2$^{WT}$ and GFP where the reported tumor incidence was 2 out of 10 mice and 3 out of 10 mice, respectively, by the end of the 10 week study period (*Berger et al., 2012*). Taken in context, the early onset of tumors in the control groups of this replication attempt effectively masks any potential effect of the *PREX2* mutations being examined. The median TFS of all groups, especially the GFP and PREX2$^{WT}$ melanocyte injected mice, were dramatically shorter than what was reported in the original study and what was anticipated in the design of this replication study (*Figure 3—figure supplement 1*).

To compare survival distributions a Cox proportional hazards regression model (CPH) was performed to better account for the multiple ties observed. As outlined in the Registered Report (*Chroscinski et al., 2014*), we planned to conduct four comparisons using the Bonferroni correction to adjust for multiple comparisons making the *a priori* Bonferroni adjusted significance threshold 0.0125 (.05/4). Although the Bonferroni method is conservative, it was accounted for in the

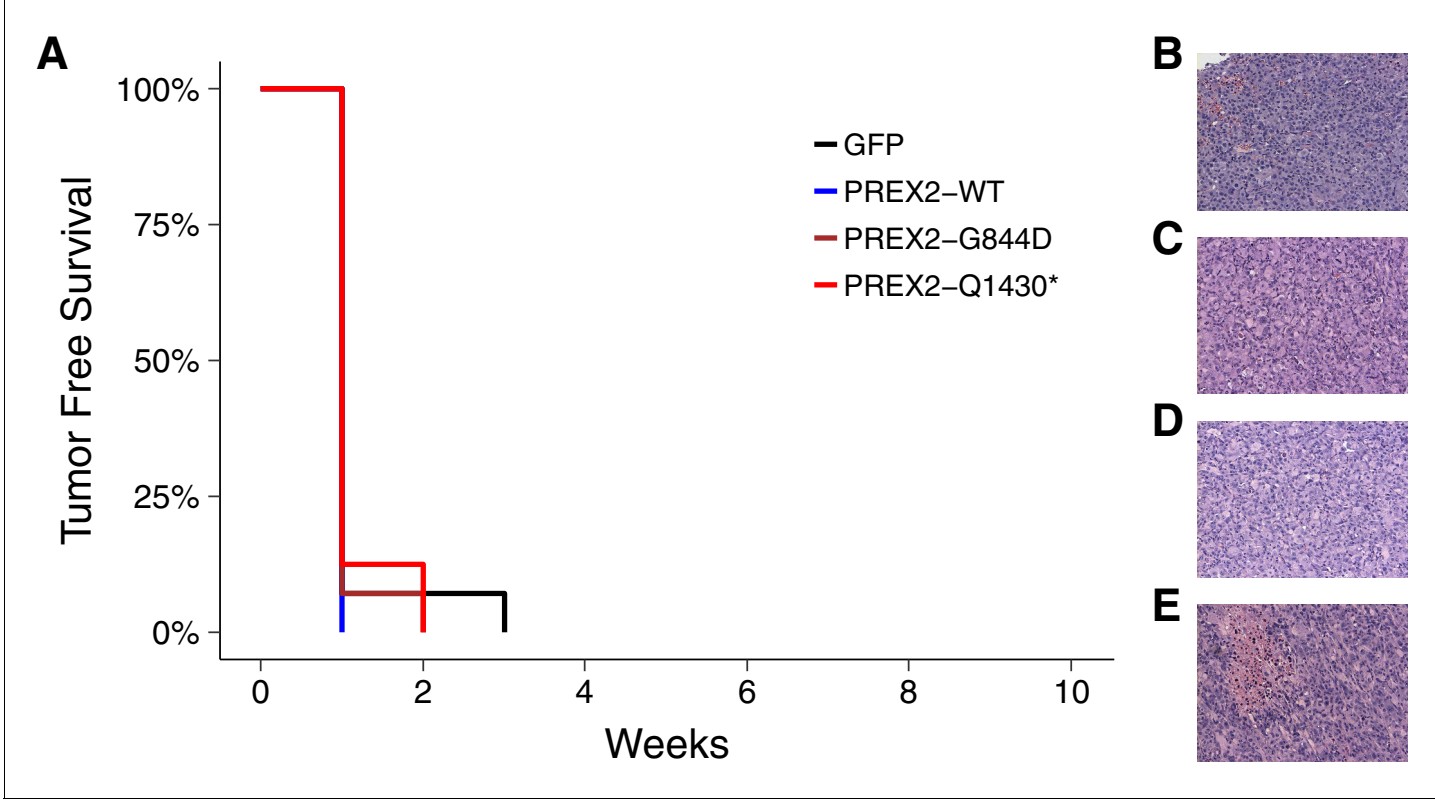

**Figure 3.** Tumor-free survival and tumor histopathology. (A) Kaplan-Meier plot of tumor-free survival (TFS). Female athymic nude mice subcutaneously injected in the right flank with NRAS$^{G12D}$ melanocytes harboring GFP [n = 14] or PREX2 variants (PREX2$^{WT}$[n = 7], PREX2$^{G844D}$[n = 14], or PREX2$^{Q1430*}$[n = 8]) were monitored every other day for tumor growth. Cox proportional hazards regression model (CPH) with *a priori* Bonferroni adjusted significance threshold = 0.0125. GFP vs. PREX2$^{G844D}$, uncorrected $p$=0.847 (Bonferroni corrected $p$>0.99); GFP vs. PREX2$^{Q1430*}$, uncorrected $p$=0.944 (Bonferroni corrected $p$>0.99); PREX2WT vs. PREX2$^{G844D}$, uncorrected $p$=0.652 (Bonferroni corrected $p$>0.99); PREX2WT vs. PREX2$^{Q1430*}$, uncorrected $p$=0.529 (Bonferroni corrected $p$>0.99). Median TFS for all groups = 1 week. (B–E) Representative histological section of (B) GFP xenograft, (C) PREX2$^{WT}$ xenograft, (D) PREX2$^{G844D}$ xenograft, and (E) PREX2$^{Q1430*}$ xenograft stained with hematoxylin and eosin. Additional details for this experiment can be found at https://osf.io/anf2s/.

The following figure supplement is available for figure 3:

**Figure supplement 1.** Table summarizing incidence rates for both Berger et al. and the current study by week after injection with NRAS$^{G12D}$ melanocytes harboring PREX2 isoforms.

power calculations to ensure sample size was sufficient. Mice injected with NRAS$^{G12D}$ melanocytes expressing GFP compared to either of the two mutant PREX2 isoforms were not statistically different (GFP vs. PREX2$^{G844D}$: uncorrected $p$=0.847, Bonferroni corrected $p$>0.99; GFP vs. PREX2$^{Q1430*}$: uncorrected $p$=0.944, Bonferroni corrected $p$>0.99), nor were mice injected with PREX2$^{WT}$ compared to either of the two mutant isoforms (PREX2$^{WT}$ vs. PREX2$^{G844D}$: uncorrected $p$=0.652, Bonferroni corrected $p$>0.99; PREX2$^{WT}$ vs. PREX2$^{Q1430*}$: uncorrected $p$=0.529, Bonferroni corrected $p$>0.99) (*Figure 3A*).

Following the detection of a palpable tumor the growth of the tumor was monitored for each mouse as specified in the Registered Report (*Chroscinski et al., 2014*). Caliper measurements were taken every seven days until tumors reached 1.5cm$^3$ or until the end of the observation period, 91 days from the first tumor measurement (*Figure 4*, *Figure 4—figure supplement 1*). There are multiple approaches that could be taken to explore this data. We determined the area under the curve (AUC) for each mouse to test if there were differences in the tumor growth rates between the different groups. Since not every tumor could be measured until the end of the observation period, we used the first six measurements to calculate AUC (*Figure 4*). Excluding the missing data from the two mice that did not survive during this period we performed a one-way ANOVA on the AUC

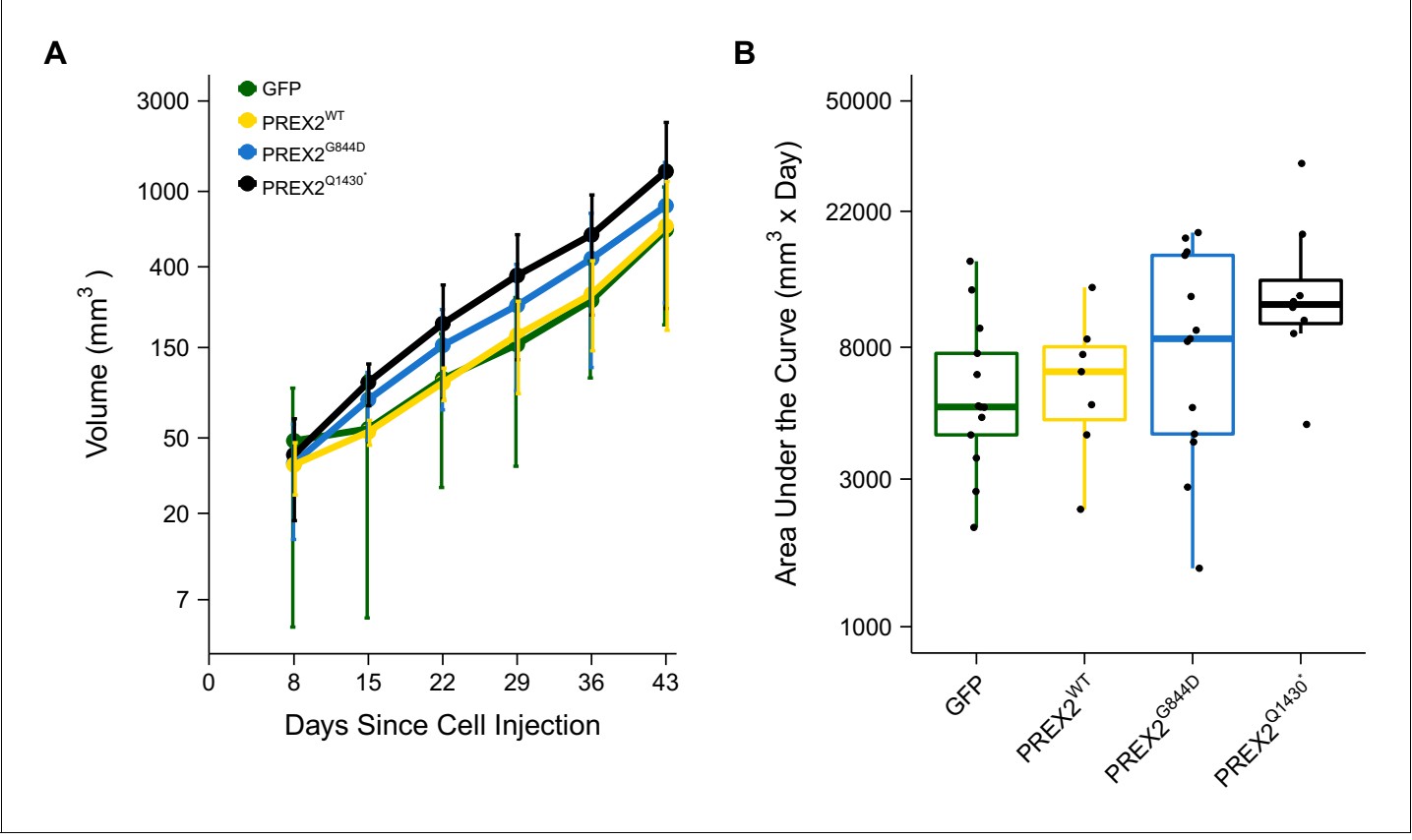

**Figure 4.** Tumor growth. Female athymic nude mice were subcutaneously injected in the right flank with NRAS[G12D] melanocytes expressing GFP or PREX2 isoforms (PREX2[WT], PREX2[G844D], or PREX2[Q1430*]). Following tumor detection, caliper measurements were taken every seven days and used to calculate tumor volume. (A) Line graph of first 6 tumor volume measurements of mice before tumors began to reach 1.5 cm$^3$ and mice were euthanized (y-axis is natural log scale). Two mice (one GFP and one PREX2[G844D]) are excluded because there were euthanized during the reported timeframe. Means reported and error bars represent SD. Number of mice: GFP [n = 13], PREX2[WT] [n = 7], PREX2[G844D] [n = 13], and PREX2[Q1430*] [n = 8]. (B) The first six tumor volume measurements were used to calculate area under the curve (AUC) for each mouse. Box and whisker plot with median represented as the line through the box, individual animal AUC values represented as dots, and whiskers representing values within 1.5 IQR of the first and third quartile (y-axis is natural log scale). One-way ANOVA on AUC (natural log-transformed); $F(3,37) = 2.411$, $p=0.0824$, $\eta_P^2 = 0.164$, 90% CI [0,0.290]. Four contrasts were performed: GFP vs. PREX2[G844D]: $t(37) = 1.217$, uncorrected $p=0.231$, Bonferroni corrected $p=0.925$, Cohen's $d = 0.477$, 95% CI [−0.308, 1.253]; GFP vs. PREX2[Q1430*]: $t(37) = 2.568$, uncorrected $p=0.014$, Bonferroni corrected $p=0.058$, Cohen's $d = 1.154$, 95% CI [0.188, 2.094]; PREX2[WT] vs. PREX2[G844D]: $t(37) = 0.752$, uncorrected $p=0.457$, Bonferroni corrected $p>0.99$, Cohen's $d = 0.352$, 95% CI [−0.578, 1.274]; PREX2[WT] vs. PREX2[Q1430*]: $t(37) = 1.988$, uncorrected $p=0.054$, Bonferroni corrected $p=0.217$, Cohen's $d = 1.029$, 95% CI [−0.076, 2.099]. Additional details for this experiment can be found at https://osf.io/anf2s/.

The following figure supplement is available for figure 4:

**Figure supplement 1.** Individual tumor xenografts.

(natural log-transformed), which was not statistically significant, $F(3,37) = 2.411$, $p=0.0824$, $\eta_P^2 = 0.164$, 90% CI [0,0.290]. Follow-up tests were performed to further explore if there were differences between the groups. The comparison between tumor growth in GFP expressing cells compared to either of the two different PREX2 isoforms were not statistically different (GFP vs. PREX2[G844D]: $t(37) = 1.217$, uncorrected $p=0.231$, Bonferroni corrected $p=0.925$, Cohen's $d = 0.477$, 95% CI [−0.308, 1.253]; GFP vs. PREX2[Q1430*]: $t(37) = 2.568$, uncorrected $p=0.014$, Bonferroni corrected $p=0.058$, Cohen's $d = 1.154$, 95% CI [0.188, 2.094]). Similarly, the comparison between tumor growth in PREX2[WT] expressing cells compared to either of the two different PREX2 isoforms were not statistically different (PREX2[WT] vs. PREX2[G844D]: $t(37) = 0.752$, uncorrected $p=0.457$, Bonferroni corrected $p>0.99$, Cohen's $d = 0.352$, 95% CI [−0.578, 1.274]; PREX2[WT] vs. PREX2[Q1430*]: $t$

(37) = 1.988, uncorrected $p$=0.054, Bonferroni corrected $p$=0.217, Cohen's $d$ = 1.029, 95% CI [−0.076, 2.099]). Interestingly, although not statistically significant, these comparisons indicate there was an increase in AUC, and thus tumor growth, in the mutant PREX2 isoforms compared to PREX2$^{WT}$ or GFP. After completion of the study, one tumor was randomly selected from each study group, sectioned, and hematoxylin and eosin (H&E) stained (*Figure 3B–E*). Histopathological examination found the tumors to be morphologically similar to each other.

## Meta-analyses of original and replicated effects

Where appropriate, we performed a meta-analysis using a random-effects model for each of the effects described above as pre-specified in the confirmatory analysis plan (*Chroscinski et al., 2014*). To provide a standardized measure of the effect, a common hazard ratio (HR) for TFS was calculated for the original and replication studies. The HR is the ratio of the probability of a particular event, in this case tumor incidence, in one group compared to the probability in another group.

The comparison of TFS distributions between mice harboring GFP or PREX2$^{G844D}$ expressing melanocytes, resulted in a HR of 4.08, 95% CI [1.07, 15.59] for the data reported in *Berger et al. (2012)*. This compares to a HR of 1.08, 95% CI [0.51, 2.29] reported in this study. A meta-analysis of these two effects resulted in a HR of 1.86, 95% CI [0.51, 6.71], $p$=0.344 (*Figure 5*). The original and replication effects were in the same direction and the point estimate of the replication effect size was within the confidence interval of the original result, while the point estimate of the original effect size was not within the confidence interval of the replication result. Further, the random effects meta-analysis did not result in a statistically significant effect.

The comparison of TFS distributions between mice harboring GFP or PREX2$^{Q1430*}$ expressing melanocytes resulted in a HR of 5.70, 95% CI [1.51, 21.62] for the data reported in *Berger et al. (2012)*. While in this study, we report a HR of 0.97, 95% CI [0.40, 2.34]. A meta-analysis of these two effects resulted in a HR of 2.19, 95% CI [0.39, 12.34], $p$=0.376 (*Figure 5*). Similarly, tests for differences in TFS distributions between PREX2$^{WT}$ and PREX2$^{G844D}$ or PREX2$^{Q1430*}$ expressing melanocytes resulted in a HR of 6.76, 95% CI [1.41, 32.41] and 9.37, 95% CI [1.96, 44.71] respectively, for the data reported in *Berger et al. (2012)*. In this study, we report a HR of 0.81, 95% CI [0.32, 2.03] for the test between PREX2$^{WT}$ and PREX2$^{G844D}$ and 0.71, 95% CI [0.25, 2.05] for the test between PREX2$^{WT}$ and PREX2$^{Q1430*}$. A meta-analysis of the PREX2$^{WT}$ and PREX2$^{G844D}$ comparisons resulted in a HR of 2.12, 95% CI [0.27, 16.84], $p$=0.477. While a meta-analysis of the PREX2$^{WT}$ and PREX2$^{Q1430*}$ comparisons resulted in a HR of 2.42, 95% CI [0.19, 30.09], $p$=0.493 (*Figure 5*). For each of these three comparisons, the original and replication results were in opposite directions and the point estimates of the replication effect sizes were not within the confidence intervals of the original results, or vice versa. Each of the random effects meta-analyses did not result in statistically significant effect. Further, the Cochran's $Q$ test for heterogeneity was statistically significant in each meta-analysis (GFP vs PREX2$^{Q1430*}$, $p$=0.030; PREX2$^{WT}$ vs PREX2$^{G844D}$, $p$=0.022; PREX2$^{WT}$ vs PREX2$^{Q1430*}$, $p$=0.007), which along with large confidence intervals around the weighted average effect sizes from the meta-analyses suggests heterogeneity between the original and replication studies.

This direct replication provides an opportunity to understand the present evidence of these effects. Any known differences, including reagents and protocol differences, were identified prior to conducting the experimental work and described in the Registered Report (*Chroscinski et al., 2014*). However, this is limited to what was obtainable from the original paper and through communication with the original authors, which means there might be particular features of the original experimental protocol that could be critical, but unidentified. So while some aspects, such as cell line, site of injection, number of cells injected, and strain and sex of mice were maintained, others were unknown or not easily controlled for. These include variables such as *PREX2* copy number in transformed cells, circadian biological responses to therapy (*Fu and Kettner, 2013*), the microbiome of recipient mice (*Macpherson and McCoy, 2015*), housing temperature in mouse facilities (*Kokolus et al., 2013*), and cell line drift (*Hughes et al., 2007*; *Kleensang et al., 2016*). Additionally, the accumulation of mutations during cell passage in vitro can drive cell lines towards a malignant phenotype that is observed in vivo (*Gregoire et al., 2001*; *Hurlin et al., 1991*). Whether these or other factors influence the outcomes of this study is open to hypothesizing and further investigation, which is facilitated by direct replications and transparent reporting.

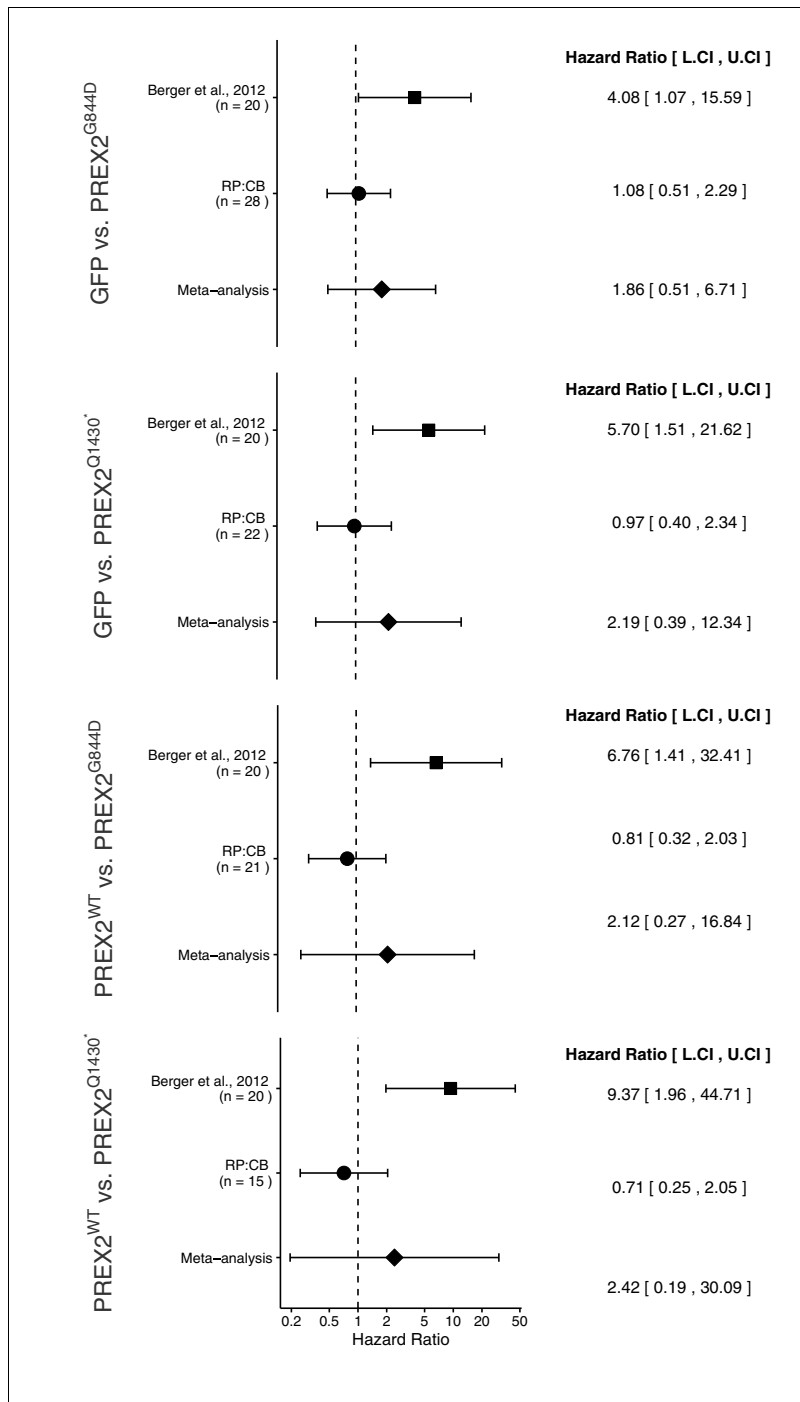

**Figure 5.** Meta-analyses of each effect. Effect size and 95% confidence interval are presented for *Berger et al. (2012)*, this replication attempt (RP:CB), and a random effects meta-analysis of the two effects. Sample sizes used in *Berger et al. (2012)* and this replication attempt are reported under the study name. Hazard Ratio (HR) of tumor-free survival in mice bearing tumors from NRAS$^{G12D}$ melanocytes expressing PREX2 isoforms or GFP are shown. GFP vs. PREX2$^{G844D}$ (meta-analysis $p$=0.344), GFP vs. PREX2$^{Q1430*}$ (meta-analysis $p$=0.376), PREX2$^{WT}$ vs. PREX2$^{G844D}$ (meta-analysis $p$=0.477), PREX2$^{WT}$ vs. PREX2$^{Q1430*}$ (meta-analysis $p$=0.493). Additional details for this experiment can be found at https://osf.io/ys8hm/.

## Materials and methods

As described in the Registered Report (*Chroscinski et al., 2014*), we attempted a replication of the experiments reported in Figures 3B and S6 of *Berger et al. (2012)*. A detailed description of all protocols can be found in the Registered Report (*Chroscinski et al., 2014*). Additional detailed experimental notes, data, and analysis are available on the Open Science Framework (OSF) (RRID:SCR_003238) (https://osf.io/jvpnw/; *Horrigan et al., 2016*).

### Cell culture

pMEL/hTERT/CDK4(R24C)/p53DD/NRAS$^{G12D}$ (NRAS$^{G12D}$) melanocytes (Dr. Yonathan Lissanu Deribe, MD Anderson Cancer Center) were maintained in Ham's F10 medium supplemented with 10% heat inactivated Fetal Bovine Serum (FBS) and 1% penicillin/streptomycin at 37°C with 5% $CO_2$. Quality control data are available at https://osf.io/38wq2/. This includes results confirming the cell lines were free of mycoplasma contamination. Additionally, STR DNA profiling of all cell lines were performed. Stable NRAS$^{G12D}$ melanocytes generated during this replication attempt were compared to the parental NRAS$^{G12D}$ melanocytes and stable NRAS$^{G12D}$ melanocytes used in the original study and provided by the original authors. All cells were confirmed to fit the same STR profile.

### Endogenous *PREX2* PCR, cDNA conversion, and sequencing

NRAS$^{G12D}$ melanocytes were used as the input for RNA isolation and subsequent cDNA generation. RNA was isolated using Qiagen RNeasy (Qiagen Cat# 74104) and Qiagen QIAshredder kits (Qiagen Cat# 79566) in addition to EtOH (Sigma-Aldrich Cat#E7023) using standard kit protocols. Next, 8 µL of RNA (1 µg) was converted to cDNA using the Superscript III First Strand Synthesis Kit (Thermo Fisher Scientific Cat# 18080051). 1 µL of random primers were mixed with 1 µl dNTPs (10 mM stock). The reaction was incubated at 65°C for 5 min and then cooled on ice. Next, 10 µL of cDNA synthesis mix was added and the reaction was incubated for 50 min at 50°C and 5 min at 85°C. RNaseH was added to the reaction and incubated for 1 hr. The sample was stored at −20°C until use.

cDNA was used as the input material to generate 37 individual PCR products for primers used to sequence endogenous PREX2 (RefSeq: NM_024870.3, GRCh38/hg38 Assembly). The PCR reaction consisted of ~47 ng cDNA (1 µL), 1 µL forward and reverse primers (10 µM) (list of primer sequences are available at https://osf.io/e6rb4/), 5 µL 10x PCR buffer, 1 µL 4x dNTPs (10 µM), 40.75 µL $H_2O$, and 0.25 µL Taq Polymerase. Two different PCR cycling conditions were used as follows: [1 cycle 95°C – 30 cycles 95°C for 30 s, 52°C for 30 s, 72°C 90 s – 1 cycle 72°C for 7 min] or [1 cycle 95°C – 40 cycles 95°C for 30 s, 52°C for 30 s, – 1 cycle 72°C for 60 s]. PCR products were run on a 1% gel to check for specificity.

Once specificity was achieved PCR products were cleaned up using Exo SAP-IT kit (Thermo Fisher Scientific Cat# 78200). The reaction was performed as follows: 10 µL PCR product and 4 µL Exo SAP-IT reagent were combined and incubated at 37°C for 15 min and then 85°C for 15 min. PCR products were sequenced using the standard Sanger method and the Big DYe Terminator 3.1 cycle sequencing kit (Thermo Fisher Scientific Cat#4337455). Standard operation procedures according to the manufacturer were followed on an ABI 3730 XL DNA analyzer (Applied Biosystems Inc., Foster City, California). Post-sequencing QC was performed and only reads with Phred scores > 20 were accepted. All sequencing reads were aligned to *PREX2* to ensure at least 2x coverage was achieved for the *PREX2* coding region.

### *PREX2* variant detection

Raw. ab1 and. seq files are available at https://osf.io/vsxed/files. As a first pass, all. ab1 files were inspected by hand for double peaks, indicating a heterozygous change with 4peaks software (http://nucleobytes.com/4peaks/). Next,. seq files were combined into one file in. fasta format (https://osf.io/2pjre/) and aligned to the human genome (GRCh38/hg38 Assembly) using NHGRI Genome Browser BLAT function and an unbiased approach (http://genome.ucsc.edu/cgi-bin/hgBlat?command=start). BLAT indicated bases that did not match the reference genome and all changes were inspected on original traces for the quality of the base call as well as double peak status. Only bases with Phred scores greater than or equal to 20 were analyzed further, which gives 99% confidence in the base call, as calculated by PeakTrace Online (https://www.nucleics.com/peaktrace-sequencing/). All changes were recorded and can be seen in *Figure 1—figure supplement 1*.

## Western blot

NRAS$^{G12D}$ melanocytes harboring stable integration of GFP, PREX2$^{WT}$, PREX2$^{G844D}$, or PREX2$^{Q1430*}$ were allowed to grow to the log phase in three separate dishes for each stable cell line. Cells were lysed on ice with lysis buffer as described in the Registered Report (*Chroscinski et al., 2014*). Cell lysates were centrifuged at 15,000 rpm at 4°C and the protein concentration of the supernatant was quantified using a BCA protein assay kit (Sigma-Aldrich Cat# BCA1) following manufacturer's instructions. 40 µg of cell lysate were loaded into a pre-cast polyacrylamide 4–12% Tris-glycine gel (Sigma-Aldrich Cat# PCG2003-10EA) along with Magic Mark XP Western Protein Standard (Thermo Fisher Scientific Cat# LC5602). The gel was run at 180V for 1 hr at RT. Protein was transferred onto a nitrocellulose membrane at RT for 2 hr at 180 mA. As a control the membrane was stained with Ponceau-S to ensure efficient transfer (https://osf.io/nm2jv/files/). The membrane was blocked with 5% milk in 1x TBS with 0.1% Tween-20 (TBS-T) overnight at 4°C on an orbital shaker. Membranes were cut at ~74 kDa to allow for parallel probing. Next, the membrane was incubated with the primary antibody anti-PREX2 (*Hodakoski et al., 2014*), made as a custom order by Zymed Laboratories, San Francisco, CA), anti-V5 (Invitrogen Cat# 451098, RRID:AB_2532221), or anti-$\alpha$-Tubulin, clone DM1A (Sigma-Aldrich Cat# T9026, RRID:AB_477593) overnight at 4°C on an orbital shaker. Primary antibodies were diluted in 5% bovine serum albumin in TBS-T containing 0.05% sodium azide at 1:10,000 for anti-PREX2, 1:5000 for anti-V5, and 1:5000 for anti-$\alpha$-Tubulin, respectively, and membranes were incubated overnight at 4°C. The anti-PREX2 antibody (Abcam Cat# Ab169027, RRID: AB_2566813) listed in the Registered Report was initially tried; however, there was no detectable PREX2 signal (image available at https://osf.io/yr6te/). Membranes were then washed and incubated with secondary antibody for 40 min at RT on an orbital shaker: Anti-rabbit IgG, HRP-linked Antibody (Cell Signaling Technologies Cat# 7074, RRID:AB_2099233); 1:5000 dilution used with anti-PREX2 antibody; Anti-mouse IgG, HRP-linked Antibody (Cell Signaling Technologies Cat# 7076; RRID:AB_ 10695470); 1:2000 dilution used with anti-V5 and anti-$\alpha$-Tubulin antibodies. Membranes were next washed and incubated with SuperSignalTM West Femto Maximum Sensitivity Substrate (Thermo Fisher Scientific Cat# 34095), according to the manufacturer's instructions. Western blots were visualized using the Chemidoc XRS imager (Bio-Rad, Hercules, California). Files were then exported as high resolution images following the manufacturer's guidelines. Raw densitometry counts for each band were obtained using ImageJ software (RRID:SCR_003070), version 1.38x (*Schneider et al., 2012*).

## Stable cell generation: Lentivirus production, cell infection, and selection

Plasmids were transformed into One Shot Stbl3 Chemically Competent *E. coli* (Invitrogen Cat# C737303) (lentiviral plasmids: GFP, PREX2$^{WT}$, PREX2$^{G844D}$, or PREX2$^{Q1430*}$; [Dr. Yonathan Lissanu Deribe, MD Anderson Cancer Center]) or One Shot TOP10 Chemically Competent *E. coli* (Invitrogen Cat# C404003) (packaging plasmids: pMD2-Gag/Pol, pMD2 VSVG, or pRSV REV; [Dr. Yonathan Lissanu Deribe, MD Anderson Cancer Center]). Clones were selected, grown in larger cultures, and DNA isolated with the GenElute Endotoxin-free Plasmid Maxiprep Kit (Sigma-Aldrich Cat# PLEX15-1KT). Plasmids were sequenced to confirm their integrity (Sequencing files can be found at https://osf.io/dhch3/). Next, HEK293T cells (ATCC Cat# CRL-3216, RRID:CVCL_0063), maintained in DMEM supplemented with 10% FBS at 37°C with 5% CO$_2$, were transfected with the appropriate lentiviral and packaging plasmids using Lipofectamine 2000 (Invitrogen Cat# 52887) and Opti-MEM (Invitrogen Cat# 31985–070) as described in the Registered Report. Two plates were transfected for each virus being produced. After 6 hr, Opti-MEM was removed from the HEK293T cells and replaced with fresh DMEM supplemented with 10% FBS at 37°C with 5% CO$_2$. After 48 hr and 72 hr post-transfection, virus was collected by removing medium and filtering through a 45 micron filter. Fractions from both days were combined and stored until used for NRAS$^{G12D}$ infection. NRAS$^{G12D}$ melanocytes were seeded at 50% confluence in 6 cm plates. After 24 hr, the medium was removed and cells were incubated with 3 mL of viral supernatant containing 8$\mu$g polybrene. After 24 hr, viral medium was removed and replaced with with fresh medium containing 5 µg/mL blasticidin. Fresh medium with blasticidin was exchanged every two days and cells were incubated for a total of 6 days. After selection with blasticidin cells were placed in fresh medium without blasticidin and expanded for use in downstream protocols.

## Animals

All animal procedures were approved by the Noble Life Sciences Inc. IACUC#5-05-002SCI (Noble Study No: S06-106) and in accordance with the Noble Institutional Animal Care and Use Committee policies on the care, welfare, and treatment of laboratory animals, which adhere to the regulations outlined in the USDA Animal Welfare Act (9 CFR Parts 1,2, and 3) and the conditions specified in the Guide for the Care and Use of Laboratory Animals (*National Research Council (US) Committee for the Update of the Guide for the Care and Use of Laboratory Animals, 2011*). For all experiments, 6–8 week old female athymic nude mice (Charles River Laboratories, Crl:NU(NCr)-Foxn1nu [Strain code 490]) were used. All mice were housed in a disposable ventilated cage, with a 12 hr light/12 hr dark cycle, at 19°C to 22°C, and 40% to 65% relative humidity. All animals were fed Harlan #2018 rodent diet and sterile acidified water *ad libitum*. Mice were monitored every day for clinical observation and every other day for signs of tumor growth. Mice were weighed once a week during the duration of the study. Animals removed from the study were anesthetized with isoflurane and sacrificed by cervical dislocation.

## Cell injection

Female athymic nude mice were anesthetized with isoflurane using a vaporizer before injection. Cells were suspended in a 1:1 ratio of HBSS buffer and High Concentration Matrigel (BD Biosciences Cat# 354262, Lot#4090005). Mice were injected with with $1 \times 10^6$ cells in the right flank in the subcutaneous space using a cold 26-gauge needle and a 0.5 mL insulin syringe.

## Tumor growth observation

Once tumor growth was detected in any animal, tumors were measured using a digital caliper and body weights recorded once per week for the duration of the study. The study group was blinded to the technician taking the measurements. Any animal determined to have a tumor that was 1.5 cm$^3$ or larger was sacrificed. Total observation continued for 91 days, post tumor detection, where possible. Four animals had not reached 1.5 cm$^3$ tumor volume at 91 days and were sacrificed at this time.

## Histopathology

When an animal was removed from the study, the animal was sacrificed, and tumor tissue was removed, cleaned of surrounding fat tissue, and placed in 10% buffered formalin. One sample was randomly selected from each study group (GFP, PREX2$^{WT}$, PREX2$^{G844D}$, and PREX2$^{Q1430*}$) and the tissue was dehydrated, placed in xylene, infiltrated, and embedded in a paraffin block. Sections were cut with a microtome to a thickness of 5 µm. Two sections of each tumor were collected on a single slide and slides were deparaffinized twice in xylene and rehydrated through graded ethanol. Slides were stained in Carazzi's hematoxylin, rinsed in water, and stained with Eosin. Slides were then treated with graded ethanol, xylene, and cover slides mounted with Permount. Histopathological reports and images can be found at https://osf.io/jvra5/.

## Statistical analysis

Statistical analysis was performed with R software (RRID:SCR_001905), version 3.3.0 (*R Core Team, 2016*). All data, csv files, and analysis scripts are available on the OSF (https://osf.io/jvpnw/). Confirmatory statistical analysis was pre-registered (https://osf.io/rvyg5/) before the experimental work began as outlined in the Registered Report (*Chroscinski et al., 2014*). Additional exploratory analysis was performed using Western blot data using the anti-PREX2 antibody and area under the curve (AUC) of tumor caliper measurement data. Data were checked to ensure assumptions of statistical tests were met. A meta-analysis of a common original and replication effect size was performed with a random effects model and the *metafor* package (*Viechtbauer, 2010*) (available at https://osf.io/ys8hm/). The original study data were shared by the original authors *a priori* during preparation of the experimental design. The data were published in the Registered Report (*Chroscinski et al., 2014*) and were used in the power calculations to determine the sample size for this study.

## Deviations from registered report

The source of pre-cast polyacrylamide 4–12% Tris-glycine gel and anti-PREX2 antibody is different than what is listed in the Registered Report, with the used source and catalog number listed above.

Additional materials and instrumentation not listed in the Registered Report, but needed during experimentation are also listed.

The survival analysis was analyzed with a COX proportional hazard (CPH) model because ties, which are prevalent in the data, are better handled with CPH than the proposed log-rank (Mantel-Cox) test.

## Acknowledgements

The Reproducibility Project: Cancer Biology would like to thank Dr. Yonathan Lissanu Deribe (MD Anderson Cancer Center) for sharing critical reagents and data, specifically the parental and stable NRAS$^{G12D}$ melanocytes and lentiviral plasmids, as well as Dr. Ramon Parsons and Dr. Douglas Barrows (Icahn School of Medicine at Mount Sinai) for sharing the anti-PREX2 antibody. We would also like to thank the following companies for generously donating reagents to the Reproducibility Project: Cancer Biology; American Type and Tissue Collection (ATCC), Applied Biological Materials, BioLegend, Charles River Laboratories, Corning Incorporated, DDC Medical, EMD Millipore, Harlan Laboratories, LI-COR Biosciences, Mirus Bio, Novus Biologicals, Sigma-Aldrich, and System Biosciences (SBI).

## Additional information

### Group author details

Reproducibility Project: Cancer Biology

Elizabeth Iorns: Science Exchange, Palo Alto, United States; Alexandria Denis: Center for Open Science, Charlottesville, United States; Stephen R Williams: Center for Open Science, Charlottesville, United States; Nicole Perfito: Science Exchange, Palo Alto, United States; Timothy M Errington, http://orcid.org/0000-0002-4959-5143: Center for Open Science, Charlottesville, United States

### Competing interests

SKH: Noble Life Sciences is a Science Exchange associated lab. PC, DS: BioFactura is a Science Exchange associated lab. FZ and SC: TACGen is a Science Exchange associated lab. RP:CB: EI, NP: Employed by and hold shares in Science Exchange Inc.

### Funding

| Funder | Author |
| --- | --- |
| Laura and John Arnold Foundation | Reproducibility Project: Cancer Biology |

The funder had no role in study design, data collection and interpretation, or the decision to submit the work for publication.

### Author contributions

SKH, PC, DS, FZ, SC, Acquisition of data, Drafting or revising the article; RP:CB, Analysis and interpretation of data, Drafting or revising the article

### Ethics

Animal experimentation: All animal procedures were approved by the Noble Life Sciences Inc. IACUC#5-05-002SCI (Noble Study No: S06-106) and in accordance with the Noble Institutional Animal Care and Use Committee policies on the care, welfare, and treatment of laboratory animals, which adhere to the regulations outlined in the USDA Animal Welfare Act (9 CFR Parts 1,2, and 3) and the conditions specified in the Guide for the Care and Use of Laboratory Animals (National Research Council (US) Committee for the Update of the Guide for the Care and Use of Laboratory Animals, 2011).

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
