## [Decision Letter]

Thank you for submitting your article "Replication Study: Melanoma genome sequencing reveals frequent PREX2 mutations" for consideration by *eLife*. Your article has been reviewed by a Reviewing Editor and the evaluation has been overseen by Charles Sawyers as the Senior Editor.

The Reviewing Editor has drafted this decision to help you prepare a revised submission.

This manuscript describes a replication study that was designed to examine the reproducibility of a study that identified a role for PREX2 in prostate cancer. The replication study appears to have been performed correctly. However, there are a number of issues that are raised by the nature of the data that was obtained:

1) Introduction, paragraph two. This text refers to the identification of Prex2 mutations by Berger et al. This presentation ignores results reported by the same authors and others that have disputed this finding – this information was presented in the Registered report and should be repeated here.

2) The rapid tumor development detected in the xenograft assays using the control melanocyte tumor cells (compared with the target study) makes the detection of accelerated tumor growth in the PREX2-expressing cells extremely difficult. This should be explicitly stated in the Abstract. This is important because the failure to replicate is compromised by this finding and should be clearly noted in the Abstract.

3) The tumor size limit is reported to be 1.5cm^3^. This is consistent with the target study. However, many IACUC approvals allow tumor growth to only 1cm^3^ prior to euthanasia. The specific IACUC approval this instance should be confirmed.

---

## [Author Response]

*This manuscript describes a replication study that was designed to examine the reproducibility of a study that identified a role for PREX2 in prostate cancer. The replication study appears to have been performed correctly. However, there are a number of issues that are raised by the nature of the data that was obtained:*

*1) Introduction, paragraph two. This text refers to the identification of Prex2 mutations by Berger et al. This presentation ignores results reported by the same authors and others that have disputed this finding – this information was presented in the Registered report and should be repeated here.*

Thank you for raising this point. We have revised the Introduction to include other papers that have not identified PREX2 as a significantly mutated gene in melanoma, including the results from the Cancer Genome Atlas Network on the genomic classification of cutaneous melanoma.

*2) The rapid tumor development detected in the xenograft assays using the control melanocyte tumor cells (compared with the target study) makes the detection of accelerated tumor growth in the PREX2-expressing cells extremely difficult. This should be explicitly stated in the Abstract. This is important because the failure to replicate is compromised by this finding and should be clearly noted in the Abstract.*

We agree and have revised the Abstract to more clearly describe this aspect of the replication in relation to the original study.

*3) The tumor size limit is reported to be 1.5cm*^3^*. This is consistent with the target study. However, many IACUC approvals allow tumor growth to only 1cm*^3^
*prior to euthanasia. The specific IACUC approval this instance should be confirmed.*

We have confirmed that for this project the IACUC (Noble Life Sciences Inc. IACUC#5-05-002SCI (Noble Study No: S06-106)) approved tumor growth to 1.5cm^3^ prior to euthanasia.